# LTβR overexpression promotes plasma cell accumulation

**Jessica A. Kotov**[1], **Ying Xu**[1], **Nicholas D. Carey**[2], **Jason G. Cyster**[1]*

**1** Howard Hughes Medical Institute and Department of Microbiology and Immunology, University of California, San Francisco, CA, United States of America, **2** Department of Medicine, University of California, San Francisco, CA, United States of America

\* jason.cyster@ucsf.edu

**Data Availability Statement:** All relevant data are within the paper and its Supporting Information files.

**Funding:** This work was supported by the Howard Hughes Medical Institute funding to J.G.C. and J.A. K. National Institutes of Health Grants (R01

## Abstract

Multiple myeloma (MM), a malignancy of plasma cells (PCs), has diverse genetic underpinnings and in rare cases these include amplification of the lymphotoxin b receptor (*Ltbr*) locus. LTβR has well defined roles in supporting lymphoid tissue development and function through actions in stromal and myeloid cells, but whether it is functional in PCs is unknown. Here we showed that *Ltbr* mRNA was upregulated in mouse PCs compared to follicular B cells, but deficiency in the receptor did not cause a reduction in PC responses to a T-dependent or T-independent immunogen. However, LTβR overexpression (OE) enhanced PC formation in vitro after LPS or anti-CD40 stimulation. In vivo, LTβR OE led to increased antigen-specific splenic and bone marrow (BM) plasma cells responses. LTβR OE PCs had increased expression of *Nfkb2* and of the NF-kB target genes *Bcl2* and *Mcl1*, factors involved in the formation of long-lived BM PCs. Our findings suggest a pathway by which *Ltbr* gene amplifications may contribute to MM development through increased NF-kB activity and induction of an anti-apoptotic transcriptional program.

## Introduction

Multiple myeloma (MM), the second most common hematologic malignancy in the world, is characterized by an overaccumulation of monoclonal plasma cells (PCs) in the bone marrow (BM). MM is a heterogeneous disease in that providing a singular treatment to a large group of MM patients may improve survival for only a subset of them. Patients differ in genetic translocations, mutations, clonality, dependence on anti-apoptotic proteins, etc. [1, 2]. Clinically, there is increasing evidence that it is important to identify the optimal treatment regimen for each subset of MM patients [3–6].

Genetic studies on MM identified abnormalities associated with constitutive activation of the NF-kB pathway in ~20% of cases [7, 8]. Signaling to NF-kB occurs via canonical and non-canonical pathways and derangements in either pathway are associated with MM. Several receptors can activate NF-kB signaling in B cells including the BCR, TLRs and TNF receptor family members with the latter being most effective in engaging the non-canonical NF-kB pathway [9]. The critical role of NF-kB signaling in MM pathogenesis is supported by the clinical success of proteasome inhibitors in patients with relapsed, refractory myeloma [10] that

AI040098, R01 AI045073 to J.G.C.; T32 AI007334 to J.A.K.) and the Life Sciences Research Foundation Postdoctoral Fellowship to J.A.K. The funders had no role in study design, data collection and analysis, decision to publish, or preparation of the manuscript.

**Competing interests:** I have read the journal's policy and the authors of this manuscript have the following competing interests: JGC is a scientific advisory board member of BeBio Pharma.

**Abbreviations:** OE, overexpression; PCs, plasma cells; BM, bone marrow; DCs, dendritic cells; qPCR, quantitative PCR; GC, germinal center; FO, follicular; ILC, innate lymphoid cell.

partly act through NF-kB inhibition [11]. As well as having cell intrinsic effects, NF-kB signaling can induce adhesion molecules such as ICAM1 and this may favor microenvironment interactions that contribute to disease pathogenesis [12, 13].

The LTβR is a TNFR family member that responds to the ligands LTα1β2 and LIGHT [14]. The best-defined role of the LTβR is in lymphoid tissue neogenesis and mice lacking this receptor or downstream components of the non-canonical NF-kB signaling pathway lack lymph nodes and have a disorganized spleen. LTβR signaling is needed within lymphoid stromal cells for expression of homeostatic chemokines and for the development of follicular dendritic cells. LTβR is also expressed in myeloid cells and is important for the homeostasis and function of certain subsets of dendritic cells (DCs), macrophages and granulocytes [15–17]. In contrast to these well-defined stromal and myeloid roles of the LTβR, whether the receptor has a function in B lineage cells has not been determined.

In a study that examined MM samples for gene amplification, rare cases of LTβR amplification were detected [8]. This finding raised questions about whether the LTβR was normally expressed or functional within B lineage cells, and whether over-expression could contribute to MM pathogenesis. Here we report that LTβR is upregulated during PC development, but it does not appear to be essential for normal PC accumulation. However, overexpression (OE) of the LTβR in mouse B cells led to enhanced PC formation in vitro and in vivo. This gain-of-function effect was ligand independent. Quantitative PCR (qPCR) analysis revealed that LTβR OE PCs had enhanced expression of *Nfkb2*, *Bcl2* and *Mcl1*. This work suggests that gene amplifications leading to LTβR OE contribute to non-canonical NF-kB signaling and thereby promote anti-apoptotic factor expression and PC survival.

## Materials and methods

### Mice, cell isolations, Abs, flow cytometry, qPCR

C57BL/6 (B6) and B6-CD45.1 mice were obtained from The Jackson Laboratory or National Cancer Institute. Blimp1-GFP, CD19-Cre[+/-], B6 *Ltbr*[-/-], and Mb1-cre[+/-] were from The Jackson Laboratory or a colony maintained at UCSF. B6 *Ltb*[-/-] and B6 *Ltbr*[fl/fl] mice were provided by Alexander Tumanov's laboratory. Mice were maintained and all experiments were performed in accordance with IACUC-approved guidelines at the Laboratory Animal Resource Center (LARC) facility at UCSF. The health and well-being of the mice were monitored daily by a qualified LARC staff member. All efforts were made to minimize and alleviate distress and suffering. Mice were anesthetized in chambers connected to isoflurane and oxygen sources prior to i.v. or i.p. injection. Mice were euthanized according to institutional guidelines involving $CO_2$ exposure followed by cervical dislocation. BM chimeras were generated by lethal irradiation with 1200 rad from a cesium irradiator (split dose separated by 3 h) followed by i.v. injection with relevant BM cells. Both male and female mice were used as donors and recipients.

PCs cells were isolated by labeling spleen or BM cells with anti-CD138 antibodies conjugated to PE or APC and applying the relevant fluorophore-specific isolation kits (Stemcell Technologies). B cells were purified from splenocytes by incubating cells with biotin-conjugated Abs specific to non-B cells followed by staining with streptavidin rapidspheres and magnetic separation (Stemcell Technologies). Blocking of the LTβR ligands LTα1β2/LIGHT was assessed through the addition of 4 ug/ml of human LTβR-Fc [18] to the in vitro culture. NP-specific B cells were labeled using NP16-PE (Biosearch Technologies) or NP18-CGG-FITC (Biosearch Technologies). The SA-PE-AF647 reagent was created by applying the AF647 labeling kit (life technologies) on SA-PE (Prozyme).

Abs used for depletion and staining were purchased from BD Biosciences, Biolegend, Invitrogen Life Technologies, and eBioscience. Fluorescently labeled cells were analyzed using an

Aurora flow cytometer (Cytek). FACS sorting was performed on an Aria (BD). Flow cytometry plots were analyzed using Flowjo (Treestar).

For qPCR, RNA was prepared from sorted cells or spleen using a RNeasy kit (Qiagen) with the following primers/probes (Integrated DNA Technologies): *Ltbr* forward primer 5′-CCAG ATGTGAGATCCAGGGC-3′, reverse primer 5′-GACCAGCGACAGCAGGATG-3′; *Nfkb2* (P100) forward primer 5′- AGCCCCTGAGACAGCTGATG-3′, reverse primer 5′-GGCCTTCACAGC CATATCGA-3′; *Bcl2* forward primer 5′- GTACCTGAACCGGCATCTGC-3′, reverse primer 5′- AAACAGAGGTCGCATGCTGG-3′; *Mcl1* forward primer 5′- TTGTAAGGACG AAACGGGACTG-3′, reverse primer 5′- TTCTGATGCCGCCTTCTAGGT-3′; and hypoxanthine phosphoribosyl transferase (*Hprt*) forward primer 5′- AGGTTGCAAGCTTGCTGG T-3′, reverse primer 5′-TGAAGTACTCATTATAGTCAAGGGCA-3′. Transcript levels were detected on an ABI 7700 sequence instrument (Taqman; PE Applied Biosystems).

## Immunizations

For NP-CGG/Alum immunizations, each mouse received an i.p. injection containing 100 ug of NP-CGG (Biosearch Technologies) in 100 ul of PBS mixed with 100 ul of Alhydrogel alum adjuvant (Invivogen). For NP-Ficoll immunizations, each mouse received an i.p. injection containing 75 ug of NP54-AECM-Ficoll (Biosearch Technologies) in 200 ul of PBS.

## Vectors, retroviral transductions and chimeras

BM retroviral transductions were performed as previously described [19] using the Platinum-E packaging cell line (Cell Biolabs) and lipofectamine 2000 (Thermo Fisher Scientific). DNA used for transfection was Empty-MSCV-Thy1.1, Ltbr-MSCV-Thy1.1 and loxp-EGFP-stop-loxp-Ltbr-MSCV-Thy1.1 vectors [19].

## In vitro culture

Isolated B cells were plated in flat bottom 24 well plates with $1x10^6$ B cells per well.

The cells were stimulated with 10 ug/ml of LPS from *Escherichia coli* (Sigma-Aldrich) or 125 ng/ml anti-CD40 (FGK45.5; Miltenyi) with 25 ng/ml IL-4 (Peprotech), 5 ng/ml IL-5 (Peprotech), 25 ng/ml IL-21 (Tonbo Biosciences) in RPMI 1640 media (Fisher) containing 10% FBS (Omega Scientific), penicillin-streptomycin, glutamax, MEM non-essential amino acids, sodium pyruvate and 2-mercaptoethanol (media supplements from Gibco unless otherwise stated) for 3 days.

## Statistical analysis

Statistical tests were performed using Prism (Graphpad) software and *p* values were obtained using two-tailed unpaired *t* tests with a 95% confidence interval. S1 File contains the values used to generate prism plots for all the figures.

# Results

## Endogenous Ltbr expression in PCs

We assessed LTβR expression by B cells through qPCR analysis of various B cell subsets. CD138+TACI+CD98+ cells consisting of CD138+Blimp1-GFP$^{INT}$ immature PCs or Blimp1-GFP$^{HI}$ mature PCs were magnetically enriched and then FACS sorted (**Fig 1A**). Multiple cell types in the spleen, including stromal cells, DCs, macrophages and granulocytes, express *Ltbr* transcripts (Immgen.org) and mRNA from total spleen tissue was used as a positive control. *Ltbr* was undetectable in follicular (FO) or germinal center (GC) B cells but was

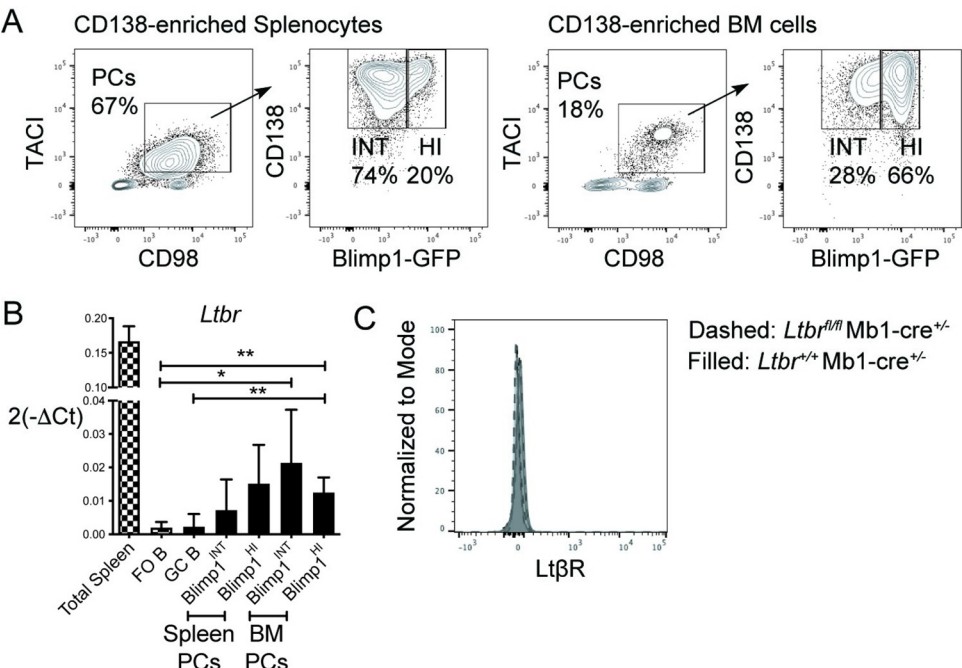

**Fig 1. LTβR expression in B lineage cells. (A)** Blimp1-GFP mice were immunized with NP-CGG/Alum, reimmunized with the same immunogen after 3 weeks and analyzed 4 days later. PCs were positively enriched from spleen and BM cells by CD138 PE magnetic bead-based selection and FACS sorted for qPCR analysis. Representative FACS plots of live Dump (CD4, CD8, F4/80, Ly6G)⁻B220⁺ cells with gates on TACI⁺CD98⁺ cells consisting of CD138⁺ Blimp1-GFP^INT immature PCs or Blimp1-GFP^HI mature PCs. **(B)** *Ltbr* gene expression relative to *Hprt* among total spleen cells, CD8⁻Gr-1⁻F4/80⁻Dead⁻ B220⁺IgD⁺ FO B cells, B220⁺PNA⁺GL7⁺IgD⁻ GC B and PCs. FO and GC B cells were FACS sorted from spleens of WT mice immunized with NP-CGG/Alum for 11 days. **(C)** LTβR protein expression by flow cytometry in LTβR-deficient (*Ltbr*^fl/fl Mb1-cre⁺/⁻) and WT (*Ltbr*^+/+ Mb1-cre⁺/⁻) PCs. Data shown in all panels are representative of two independent experiments. Pooled data from two independent experiments are shown (n = 4–6 mice per group for A-B and n = 3 for C). * $p < 0.05$, ** $p < 0.01$.

upregulated in Blimp1-GFP^INT immature and Blimp1-GFP^HI mature PCs in spleen and BM (**Fig 1B**). In accord with the low transcript abundance and the low sensitivity of flow cytometric staining compared to qPCR, we were unable to detect LTβR protein on the surface of PCs (**Fig 1C**).

## Intact PC responses by LTβR-deficient B cells to T cell-dependent and -independent immunogens

The role of LTβR in the B lineage was explored by analyzing chimeras containing CD45.2⁺ C57BL/6J or LTβR⁻/⁻ BM cells mixed with CD45.1⁺ wild-type BM cells at a 30:70 ratio, in CD45.1⁺ recipients. After 8 weeks of reconstitution, the chimeric mice were immunized with the T cell-dependent antigen NP-CGG/Alum two times, 21 days apart, and analyzed 28 days after the booster immunization. A late timepoint following an antigen booster immunization served the purposes of examining the potential survival benefits of LTβR on long-lived spleen and BM PCs as well as examining a larger number of antigen-specific BM PCs. Chimerism was assessed by comparing the CD45.2 and CD45.1 frequencies by flow cytometry analysis of the B220-expressing FO B cell populations (**Fig 2A and 2B, top**). The influence of LTβR on total and NP⁺ GC B cells and memory B cells was examined by comparing the CD45.2 and CD45.1 ratio against that of the starting FO B cells (**Fig 2A and 2B, middle-bottom**). Given the low to undetectable expression of *Ltbr* in GC B cells (**Fig 1B**) and memory B cells

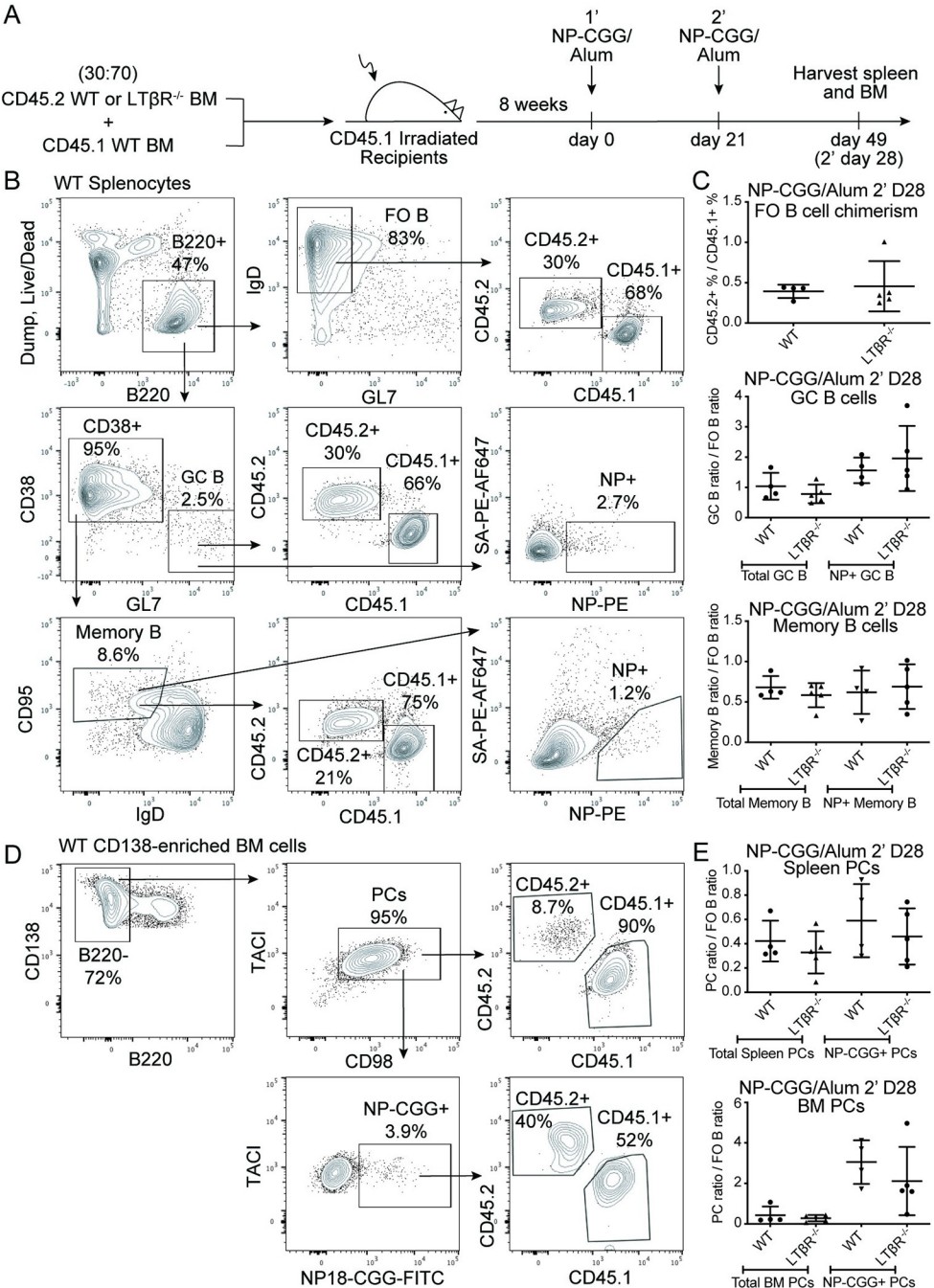

**Fig 2. LTβR expression in B cells is not required for PC responses to a T-dependent antigen. (A)** Diagram of experimental scheme. Chimeras were made by mixing CD45.2$^+$ WT or *Ltbr*$^{-/-}$ BM cells with CD45.1$^+$ wild-type cells at a 30:70 ratio and injecting into irradiated CD45.1$^+$ recipients. After 8 weeks of reconstitution, mice were immunized with NP-CGG/Alum two times 21 days apart and analyzed 28 days after the booster immunization. **(B)** FACS plots of singlet cells with a gate on live Dump$^-$B220$^+$ cells (top row, left). The B220$^+$ cells were further gated on IgD$^+$GL7$^-$ FO B cells (top row, middle), with further gates on CD45.2$^+$ or CD45.1$^+$ cells (top row, right). Alternatively, B220$^+$ cells were gated based on CD38$^-$GL7$^+$ GC B cells (middle row, left) with further gates on CD45.2$^+$ or CD45.1$^+$ cells (middle row, middle). NP$^+$ GC B cells were identified as NP16-PE$^+$ and decoy SA-PE-AF647$^-$ (middle row, right). Memory B cells were identified as CD38$^+$GL7$^-$CD95$^+$IgD$^-$ (bottom row, left) with gates on CD45.2$^+$ or CD45.1$^+$ cells (bottom row, middle) or NP$^+$ memory B cells (bottom row, right). **(C)** Chimerism based on the ratio of CD45.2$^+$ WT or *Ltbr*$^{-/-}$ to CD45.1$^+$ WT FO B cells (top). GC B cell ratio of CD45.2$^+$ cells to CD45.1$^+$ cells normalized to the FO B cell ratio (middle). Memory B cell ratio of CD45.2$^+$ cells to CD45.1$^+$ cells normalized to the FO B cell ratio (bottom). **(D)** Representative FACS plots of CD138-enriched BM cells from a CD45.2$^+$ WT recipient chimera pre-gated on live

Dump⁻B220<sup>lo</sup>CD138⁺ cells. PCs were identified as B220⁻ (left), TACI⁺CD98⁺ (top row, middle) with gates on CD45.2⁺ or CD45.1⁺ cells (top row, right). NP-CGG⁺ PCs were identified as NP-CGG-FITC⁺ (bottom row, left) with further gates on CD45.2⁺ or CD45.1⁺ cells (bottom row, right). **(E)** Ratio of CD45.2⁺ cells to CD45.1⁺ cells normalized to the FO B cell ratio for spleen PCs (top) or BM PCs (bottom). Data shown in all panels are representative of two independent experiments (n = 4–5 mice per group).

(Immgen.org), it was as expected for LTβR to not play a role in the development of these B cell subsets. The potential role of the LTβR in the PC compartment was assessed by analysis of total and NP⁺ PCs in the spleen and BM. LTβR deficiency did not cause a significant change in PC frequencies in either organ (**Fig 2C and 2D**).

We next asked whether LTβR expression impacts PCs induced by a T cell-independent immunogen. This was examined by immunizing *Ltbr*<sup>fl/fl</sup> Mb1-cre<sup>+/-</sup> or control (*Ltbr*<sup>fl/fl</sup> or *Ltbr*<sup>fl/+</sup> Mb1-cre<sup>+/-</sup>) mice with NP-Ficoll and examining the total and NP⁺ PCs in the spleen by flow cytometry after 11 days (**Fig 3A**). Quantification of the PCs in these B cell specific LTβR-deficient mice indicated a lack of importance for B lineage LTβR expression on early T cell independent PCs (**Fig 3B**). These data indicate LTβR is not critical for PC accumulation under the T-dependent or T-independent immunization conditions tested.

## LTβR OE results in greater PC accumulation after in vitro stimulation independently of ligand

Given the cases of *Ltbr* amplification found in MM patients, it seemed possible that LTβR overactivation promotes PC accumulation or survival. This hypothesis was tested by creating BM chimeras that OE LTβR specifically in the CD19⁺ B cell compartment. Chimeras were made by retroviral transduction of CD45.2⁺ CD19-Cre<sup>+/-</sup> BM with Empty-MSCV-Thy1.1 control or loxp-EGFP-stop-loxp-Ltbr-MSCV-Thy1.1 retrovirus prior to transfer into irradiated CD45.1⁺ host mice. Expression of Cre recombinase by CD19⁺ cells leads to excision of the loxp-EGFP-STOP-loxp sequence and subsequent expression of the downstream *Ltbr* gene.

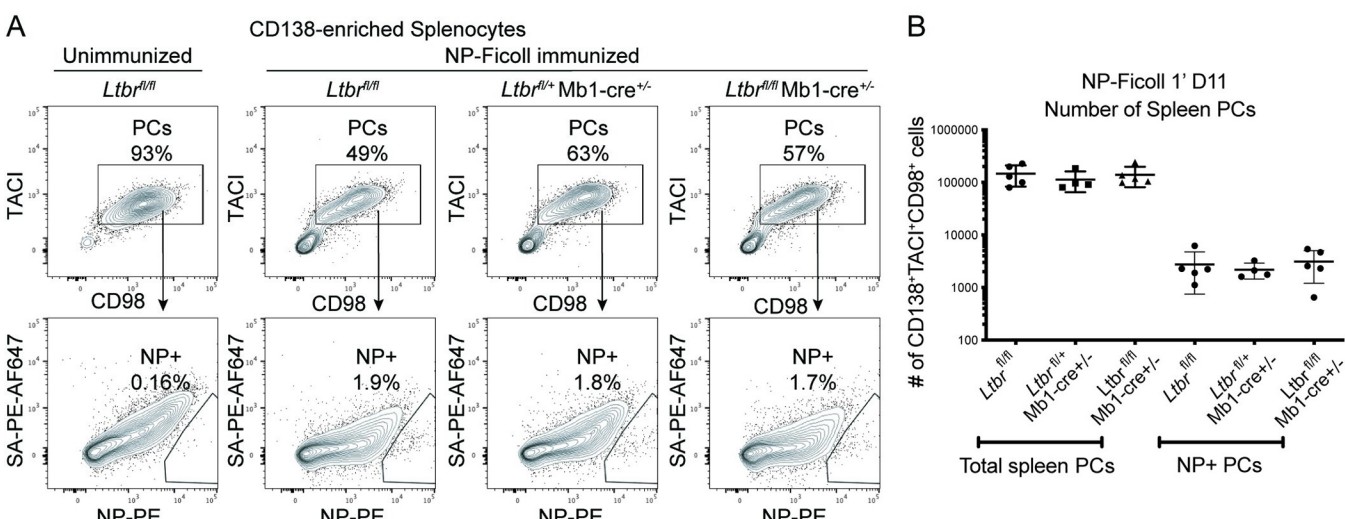

**Fig 3. LTβR expression in B cells is not required for PC responses to a T-independent antigen.** *Ltbr*<sup>fl/fl</sup> Mb1-cre<sup>+/-</sup> or control (*Ltbr*<sup>fl/fl</sup> or *Ltbr*<sup>fl/+</sup> Mb1-cre<sup>+/-</sup>) mice were immunized with NP-Ficoll for 11 days. **(A)** Representative FACS plots of live Dump⁻ CD138-enriched splenocytes from an unimmunized *Ltbr*<sup>fl/fl</sup> mouse and immunized mice (top row) with gates on TACI⁺CD98⁺ PCs. NP⁺ PCs cells were identified as NP16-PE⁺ and decoy SA-PE-AF647⁻ (bottom row). **(B)** Numeration of total and NP⁺ PCs in spleens of immunized mice. Data shown in all panels are representative of two independent experiments (n = 4–5 mice per group).

Therefore, Thy1.1$^+$ non-B cells that do no express CD19 in loxp-EGFP-stop-loxp-Ltbr-MSCV-Thy1.1 chimeric mice co-express eGFP while Thy1.1$^+$ non-B cells in Empty-MSCV-Thy1.1 chimeric mice do not (**Fig 4A**).

B cells were isolated from spleens and in vitro activated for 3 days with stimuli that mimic T cell-independent (LPS) or T cell-dependent (anti-CD40 and cytokines) responses. Transduced B cells OE LTβR were identified as Thy1.1$^+$eGFP$^-$ (**Fig 4B**). The ability of LTβR OE Thy1.1$^+$ B cells or control Thy1.1$^+$ B cells to form CD98$^+$CD138$^+$ PCs was quantified. OE of LTβR strongly enhanced the ability of B cells to form PCs after both LPS and aCD40 stimulations (**Fig 4B and 4C, left and middle**). This PC enhancement was not at the expense of Thy1.1$^-$ cells in the LPS condition, but was in the aCD40 condition (**Fig 4C, right**). It is possible that the remarkably strong augmentation of PC accumulation caused by LTβR OE in the aCD40 culture condition led to depletion of factors needed for accumulation of wild type (WT) PCs.

LTβR OE as a result of retroviral transduction was confirmed by detecting LTβR protein expression in transduced B cells using flow cytometric analysis (**Fig 4D**). Whether LTβR ligand was required for LTβR OE to enhance PC accumulation was tested by blocking any LTβR ligand that may have been available from cells in the culture using soluble LTβR-Fc [20]. Inclusion of this antagonist did not impact PC formation by LTβR OE cells (**Fig 4C**) indicating that LTβR OE leads to greater PC differentiation independently of LTβR ligands. This was further supported by the finding that LTβ-deficient B cells, which are unable to form the LTβR ligand LTα1β2 on the surface, fully retained the ability to form increased frequencies of PCs under conditions of LTβR OE (**Fig 4E**).

## LTβR OE promotes antigen-specific PC formation in the spleen and BM in response to immunization

The influence of LTβR OE on PC frequencies was assessed in vivo in retroviral BM chimeras containing control or LTβR OE B cells that were identified based on donor CD45.2 expression and the transduction marker Thy1.1. The mice were immunized with NP-CGG/Alum two times 21 days apart and after 26 days from the booster immunization, spleens and BM were analyzed. Efficiency of the retroviral transduction was assessed by quantifying the Thy1.1 frequency among the FO B cell population, which ranged from 3–56% (**Fig 5A and 5B**). Thy1.1 reporter+ cells were represented in the GC compartment at similar frequencies as in the FO compartment, suggesting that LTβR expression does not influence the GC response (**Fig 5C**). Assessment of the numbers of total PCs and NP$^+$ PCs in the spleen and BM showed that all the mice mounted similar overall responses (**Fig 5D–5F**). The influence of LTβR on PC accumulation was assessed by comparing the Thy1.1 frequency of PCs to the starting Thy1.1 frequency of FO B cells in each mouse (**Fig 5G**). Importantly, total and NP$^+$ LTβR OE PCs strongly over accumulated in the spleen and BM (**Fig 5G**) indicating that elevated LTβR expression is sufficient to augment PC accumulation in vivo.

## LTβR OE PCs have increased transcription of NF-kB and anti-apoptotic genes

We hypothesized that LTβR OE of B cells was leading to the overaccumulation of PCs in vivo at least in part by promoting a transcriptional program that prevents cell death. To test this possibility, control and LTβR OE spleen PCs from retroviral BM chimeras were FACS sorted as in **Fig 1** and analyzed for transcription of known PC-related pro-survival genes by qPCR analysis. As expected, LTβR OE PCs exhibited elevated transcription of *Nfkb2*, a non-canonical NF-kB related gene that is induced by NF-kB (**Fig 6**). Furthermore, OE of LTβR led to elevated transcription of the NF-kB target genes *Bcl2* and *Mcl1* (**Fig 6**), anti-apoptotic factors involved

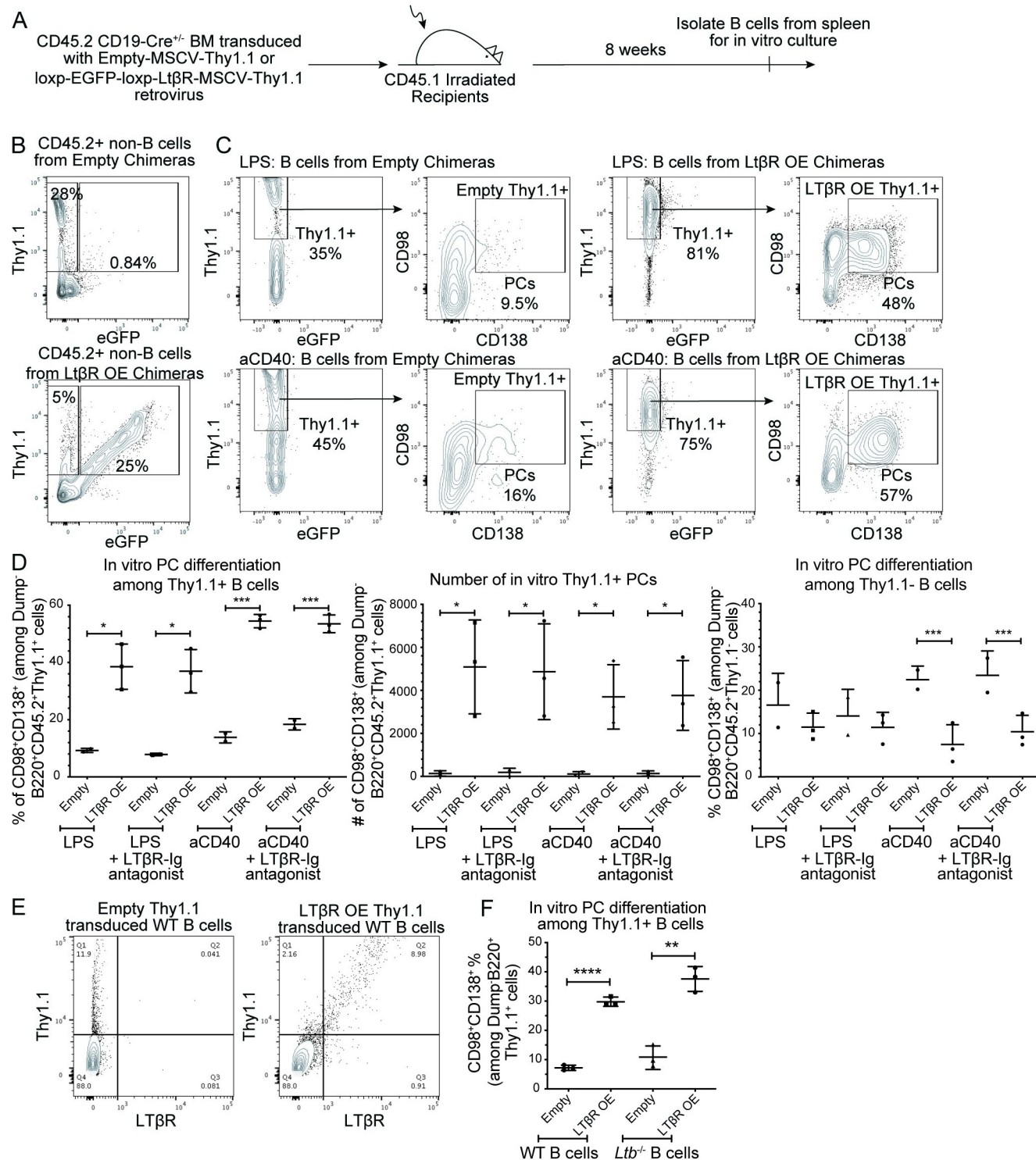

**Fig 4. OE of LTβR increases PC numbers in vitro in response to LPS or anti-CD40 stimulation in the absence of ligand. (A)** Diagram of experimental scheme. Chimeras were made by retroviral transduction of CD45.2[+] CD19-Cre[+/-] BM with Empty-MSCV-Thy1.1 or loxp-EGFP-loxp-LTβR-MSCV-Thy1.1 retrovirus and transfer into irradiated CD45.1[+] recipients. After reconstitution, B cells were isolated from the spleen for in vitro culture. **(B)** Representative FACS plots of CD45.2[+]B220[-] splenocytes in chimeras stimulated with or anti-CD40 with IL-4, IL-5, and IL-21 for 3 days. Transduction efficiency based on Thy1.1 expression on FO B cells was 43–93% among the mice analyzed. Blocking of the LTβR ligands was assessed through the addition of human LTβR-Fc blocking antibody. **(C)** FACS plots of live Dump[-]B220[+]CD45.2[+] B cells with gates on Thy1.1[+] cells and further gates on CD98[+]CD138[+] PCs. **(D)** Percentages of CD98[+]CD138[+] PCs among Thy1.1[+] (left) and Thy1.1[-] (right) B cells as gated in B. Numbers of CD98[+]CD138[+] Thy1.1[+] PCs (middle). **(E)**

LTβR protein expression by flow cytometry of WT B cells stimulated with LPS for 3 days and retrovirally transduced with Empty-MSCV-Thy1.1 or LTβR-MSCV-Thy1.1 vectors. **(F)** Frequency of CD98+CD138+ PCs among Thy1.1+ transduced WT or *Ltb*-/- B cells stimulated with LPS media for 3 days. Data shown in all panels are representative of two independent experiments (n = 2–3 mice per group). * $p < 0.05$, ** $p < 0.01$, *** $p < 0.001$, **** $p < 0.0001$.

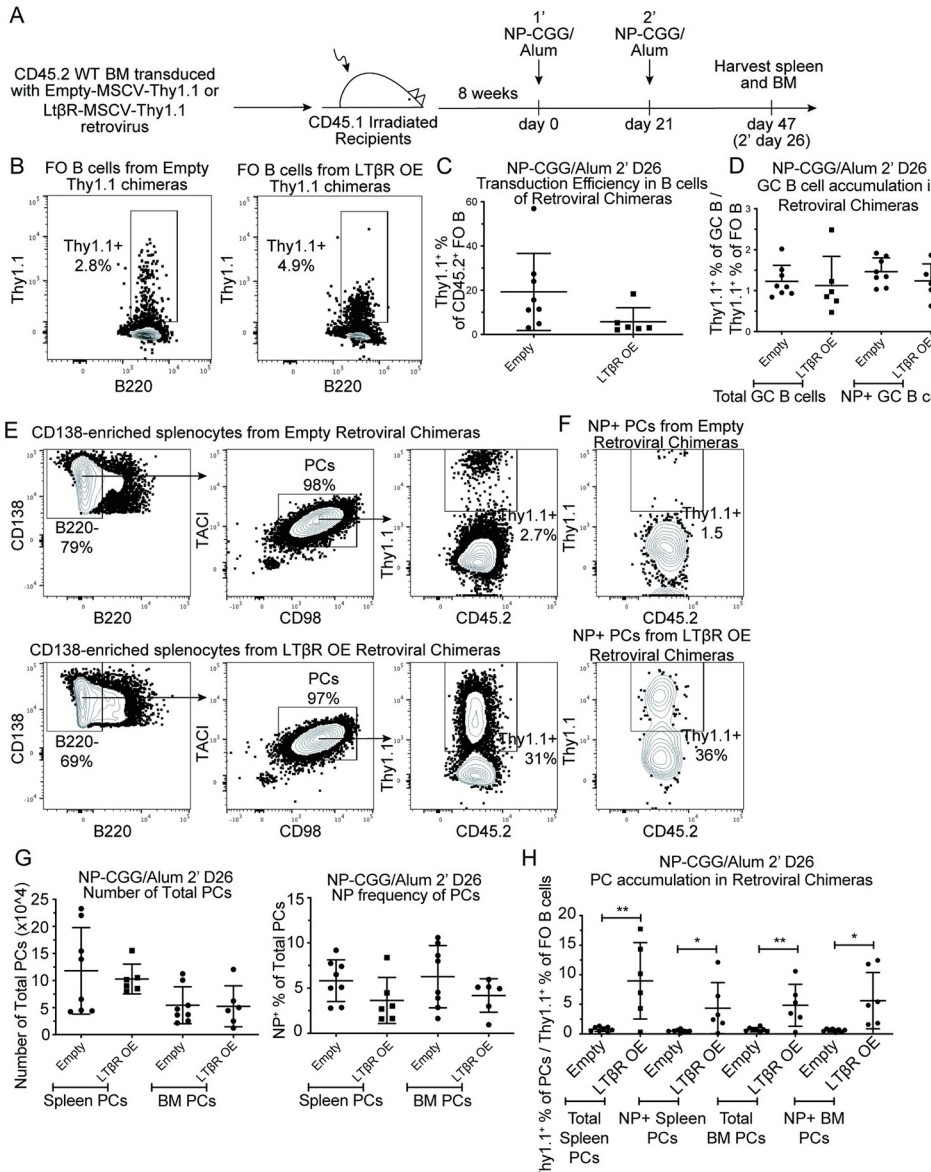

**Fig 5. LTβR OE promotes PC accumulation in vivo. (A)** Diagram of experimental scheme. Chimeras were made by retroviral transduction of CD45.2+ WT BM with Empty-MSCV-Thy1.1 or LTβR-MSCV-Thy1.1 (LTβR OE) retrovirus and transfer into irradiated CD45.1+ recipients. After reconstitution, mice were immunized with NP-CGG/Alum two times 21 days apart. After 26 days from the booster immunization, spleen and BM were analyzed. **(B)** Representative flow plots representing Dump-B220+CD45.2+CD45.1-IgD+ spleen FO B cells with gates on Thy1.1+ cells. **(C)** Thy1.1 frequency of FO B cells. **(D)** Thy1.1 frequency of spleen GC B cells normalized to Thy1.1 frequency of FO B cells in matched mice. **(E)** FACS plots of Dump-CD138+CD45.2+ cells with gates on B220-, TACI+CD98+ PCs with further gates on Thy1.1 cells among total spleen PCs or **(F)** NP+ spleen PCs. NP+ PCs were identified as NP16-PE+ and decoy SA-PE-AF647-. **(G)** Number of total spleen or BM PCs (left) and NP frequency of PCs (right). **(H)** Thy1.1 frequency of spleen or BM PCs normalized to Thy1.1 frequency of FO B cells in matched mice. Data shown in all panels are representative of two independent experiments (n = 6–8 mice per group). * $p < 0.05$, ** $p < 0.01$.

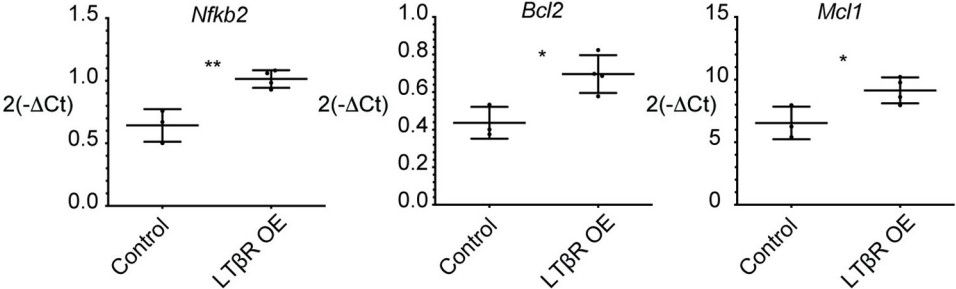

**Fig 6. LTβR OE PCs have greater transcription of *Nfkb2*, *Bcl2*, and *Mcl1*.** B cell specific LTβR OE or control chimeras were generated as in **Fig 4**. Dump⁻CD138⁺B220⁻TACI⁺CD98⁺CD45.2⁺ Thy1.1⁺ or Thy1.1⁻ PCs were sorted from spleens. qPCR analysis was conducted to determine the transcription levels of *Nfkb2*, *Bcl2*, and *Mcl1*. Control samples consist of Thy1.1⁺ or Thy1.1⁻ PCs from Empty-MSCV-Thy1.1 chimeras and Thy1.1⁻ PCs from loxp-EGFP-loxp-Ltbr-MSCV-Thy1.1 chimeras. In some instances, cells from multiple mice were combined into one sample to obtain sufficient cells for analysis. Pooled data from two independent experiments are shown. * $p < 0.05$, ** $p < 0.01$.

in the formation and maintenance of long-lived BM PCs [21, 22]. Our findings suggest a role for LTβR in enhancing non-canonical NF-kB activity and induction of an anti-apoptotic transcriptional program when overexpressed in PCs.

## Discussion

Our work establishes that *Ltbr* mRNA is upregulated in PCs compared to FO B cells. Although the expression is low in normal PCs, the locus activity is likely important for enabling high expression of the LTβR in pre-malignant PCs on occasions when locus amplification occurs. Our mouse modeling work shows that elevated LTβR expression in B lineage cells is sufficient to promote elevated PC expression of anti-apoptotic genes and increased PC accumulation (**Fig 7**). The findings are consistent with the conclusion that *Ltbr* amplifications are one of the mechanisms by which NF-kB signaling can become deregulated in PCs, contributing to their accumulation and malignant progression.

A recent genome wide OE screen identified *LTBR* as the gene that was most potent in promoting human T cell effector function and preventing exhaustion [23]. Taking a similar cell engineering perspective, our data suggest that increasing LTβR expression in engineered B lineage cells may be an approach to augment the generation of plasma cells and the associated production of antibody or introduced secretory proteins.

Our findings indicate that LTβR OE can promote PC accumulation in a ligand-independent manner. In accord with this result, OE of the LTβR in HEK293 cells that lack expression of LTα1β2 or LIGHT was sufficient to activate the NF-kB pathway [24]. These data do not exclude the possibility that LTβR OE cells signal more strongly in the presence of ligand. Multiple cell types in mouse and human BM express LTα1β2 and *Tnfrsf14* (LIGHT) (Immgen.org; HCA BM viewer). Moreover, there is evidence for LTβR signaling in the BM acting to influence granulocyte metabolism [25], HSC maintenance [26], and mesenchymal cells [27]. As well as LTβR, MM can strongly OE CD40 and TACI (TNFRSF13B) [7, 8]. Our findings add support to the possibility that these additional TNFR family members signal constitutively in PC when OE, thereby contributing to malignant progression. In the case of the LTβR, ligand independent signaling may limit the possible efficacy of LTβR antagonism as an approach to treat LTβR OE MM. Instead, better therapeutic effect may be achieved with inhibitors of downstream components such as NF-kB.

Our observation that LTβR OE promotes increased PC transcription of both the non-canonical NF-kB related gene *Nfkb2* and the anti-apoptotic gene *Bcl2* is in agreement with

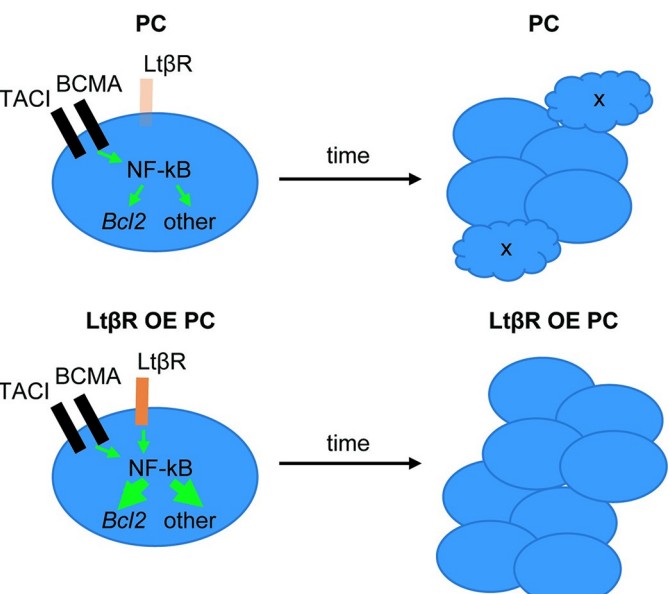

**Fig 7. Model for LTβR over-expression (OE) effect on PC accumulation.** In WT PCs, TACI and BCMA are highly expressed while LTβR expression is very low. Under conditions where LTβR is OE, there is an overall increase in signaling via NF-kB leading to augmented Bcl2-family protein expression and reduced apoptosis, and to increases in other NF-kB target genes that augment PC accumulation. While TACI and BCMA signaling is ligand (BAFF, APRIL) dependent, the OE LTβR may signal in a ligand independent manner. Chronic LTβR OE in PCs due to gene amplification may allow marked over accumulation of PCs, increasing the chances of additional genetic variants emerging and progression to MM.

studies indicating that p52 (encoded by *Nfkb2*) drives *Bcl2* transcription in mouse splenic B cells [28]. In addition, there are NF-kB binding sites in the BCL-2 promoter that can be bound and transactivated by p50 or p52 homodimers [29]. Furthermore, OE of NF-kB2-encoded p100 and its processed product p52 can induce BCL-2 expression [29]. Mcl-1 is another anti-apoptotic protein that plays a significant role in the pathogenesis of various human tumors including MM [30]. Mcl-1 induction has been linked to the canonical NF-kB pathway [31]. Furthermore, normal PCs depend on Mcl-1 for maintenance in the BM and TNFRSF17 (BCMA) signals via NF-kB to promote *Mcl1* expression [22]. These studies support the conclusion that LTbR signaling via NF-kB promotes PC accumulation in part through induction of *Bcl2* and *Mcl1*. It is likely that additional pathways not examined here also contribute to LTbR OE-induced PC accumulation. These data also point to the possible value of testing Bcl2 and Mcl1 antagonists in the treatment of LTβR OE MM [32].

A limitation in our OE studies is that *Ltbr* was over-expressed in all B lineage cells and it is possible that PC accumulation was secondary to effects of the receptor in an earlier stage of B cell development. However, we did not observe any effect of *Ltbr* OE on the fraction of B cells in the GC versus FO compartment, providing one line of evidence that receptor expression was not causing marked changes in all B cell compartments. Moreover, it is possible that the *Ltbr* amplifications observed in MM occur during the gene rearrangements events associated with B cell development rather than selectively in at the PC stage. Our findings suggest that even when gene amplifications occur early in the B lineage, the effect may not manifest until the PC stage.

The LTβR is broadly expressed by myeloid cells (Immgen.org) and we speculate that down-regulation of the B lineage-defining transcription factor Pax5 during PC differentiation [33] allows de-repression of the LTβR locus. Conditional removal of Pax5 from mature B cells

allows their reverse differentiation toward the myeloid lineage [34]. The loss of Pax5 expression leads to profound changes in locus accessibility in B cells and this presumably also occurs during the natural repression of Pax5 in PCs, but it is unclear what fraction of the de-repressed genes are of physiological importance in PCs. Although our studies have not identified a role for LTβR expression in normal PC accumulation, they do not exclude an influence of this receptor under some conditions such as in promoting PC longevity over periods of months to years or contributing to survival in inflammatory niches that may be high in LTα1β2 and LIGHT compared to BCMA and TACI ligands.

## Supporting information

**S1 File. This supporting information file includes the values used to generate prism plots for all the figures.**
(XLSX)

## Acknowledgments

The authors thank Alexander Tumanov for sharing *Ltb*$^{-/-}$ and *Ltbr*$^{fl/fl}$ mice, Jinping An for help with mouse genotyping, and the Cyster lab members for input on this project.

## Author Contributions

**Conceptualization:** Jessica A. Kotov, Jason G. Cyster.

**Data curation:** Jessica A. Kotov, Ying Xu.

**Formal analysis:** Jessica A. Kotov.

**Funding acquisition:** Jessica A. Kotov.

**Investigation:** Jessica A. Kotov, Jason G. Cyster.

**Methodology:** Jessica A. Kotov, Nicholas D. Carey.

**Supervision:** Jason G. Cyster.

**Validation:** Nicholas D. Carey.

**Visualization:** Jessica A. Kotov.

**Writing – original draft:** Jessica A. Kotov.

**Writing – review & editing:** Jason G. Cyster.

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
