## [Decision Letter · Decision Letter 0]

25 Apr 2022

PONE-D-22-01475LTβR overexpression promotes plasma cell accumulationPLOS ONE

Dear Dr. Cyster,

Thank you for submitting your manuscript to PLOS ONE. After careful consideration, we feel that it has merit but does not fully meet PLOS ONE’s publication criteria as it currently stands. Therefore, we invite you to submit a revised version of the manuscript that addresses the points raised during the review process.

We look forward to receiving your revised manuscript.

Kind regards,

Antonio Solimando

Academic Editor

PLOS ONE

Journal Requirements:

In your cover letter, please note whether your blot/gel image data are in Supporting Information or posted at a public data repository, provide the repository URL if relevant, and provide specific details as to which raw blot/gel images, if any, are not available. Email us at plosone@plos.org if you have any questions

3. To comply with PLOS ONE submissions requirements, in your Methods section, please provide additional information on the animal research and ensure you have included details on (1) methods of sacrifice, (2) methods of anesthesia and/or analgesia, and (3) efforts to alleviate suffering

Additional Editor Comments (if provided):

The authors are required to carefully answer to the reviewers' comments by providing a rebuttal point by point letter and improve the manuscript accordingly.

Reviewers' comments:

Reviewer's Responses to Questions

**Comments to the Author**

1. Is the manuscript technically sound, and do the data support the conclusions?

Reviewer #1: Yes

Reviewer #2: Yes

2. Has the statistical analysis been performed appropriately and rigorously? 

Reviewer #1: Yes

Reviewer #2: Yes

3. Have the authors made all data underlying the findings in their manuscript fully available?

Reviewer #1: Yes

Reviewer #2: Yes

4. Is the manuscript presented in an intelligible fashion and written in standard English?

Reviewer #1: Yes

Reviewer #2: Yes

5. Review Comments to the Author

Reviewer #1: The authors uncovered LTβR overexpression to potentially promote plasma cell accumulation.

Point to be considered:

1. While performing FACS analyses, did the authors employ unstained/isotype controls for compensation?

2. A graphical abstract/figure or panel wummarizing the experimental approach would be beneficial for the reader.

3.When discussing the methodology applied for number selection it is not clear for my understanding, how the author identify or discuss methodologically the experimental in vivo setting. Indeed, in the in vivo experiments, sample size should be calculated in a rigorous way, i.e. by using G*Power software (power of for example 80% and 0.05 statistical level, etc.). For example, assuming an effect-size of for example, 0.4 with statistical significance of α <;0.05 and a power of 80% a given number of 9 mice for each group for a total of 18 mice would be extimated. This number should then be increased to 20 considering an expected drop-out rate of 10% for the treatment. Can the authors comment on this?

4. Cell adhesion molecules, such as ICAM1 and JAMs, are enhanced by tumor cells in a NF-κB-dependent manner in myeloma and its microenvironment: can the authors expand this issue in the introduction/discussion section in order to boost the general interest for a broad readership from the oncology field? Indeed, As is now well known, tumors grow and evolve through a constant crosstalk with the surrounding microenvironment, and emerging evidence indicates that angiogenesis and immunosuppression frequently occur simultaneously in response to this crosstalk. Accordingly, strategies combining anti-angiogenic therapy and immunotherapy seem to have the potential to tip the balance of the tumor microenvironment and improve treatment response (refer to PMID: 32354870 and PMID: 32064051)

Reviewer #2: The authors provide intriguing insights regarding a partially neglected topic in MM, regarding the lymphotoxin b receptor (Ltbr) locus. LTβR has well defined roles in supporting lymphoid tissue development and function through actions in stromal and myeloid cells.

Point to be addressed:

1. Are the presented data following a normal (Gaussian) distribution? If this is the case, this should be clearly stated, and parametric tests can be performed. If this is not the case, non parametric tests should be performe instead

2. I would suggest to better highlight the potential limitations of this study (i.e. how would the authors translate their findings to the clinical side? The underlying message here is that more precision and pinpoint approaches needed to be tested in well designed clinical trials – a challenge, but I would be interested in their perspective of how this might be done).

3. To my knowledge, the role of endothelial lymphotoxin beta receptor (LTβR) has been described to egress lymphoid organ (in this regard endothelial cells can be important as well as MM angiogenesis) and ruled out the role of LTβR from epithelial cells or dendritic cells. In this regard, BST-1, like CD38, behaves both as an ectoenzyme and signaling receptor and has been reported to regulate the trafficking of neutrophil and monocytes. Did the authors check for this immune-microenvironmental characteristic or could they envision next step aiming to compensate for this in the next future?

4. In the frame of point 3 thinking, CD38/LTβR could be relevant, given the importance of CD38 targeting in MM: CD31/CD38 axis activates genetic programs relevant for proliferative responses. It also indicates a contribution of this pathway to the processes mediating migration and homing. These results further support the notion that the CD31/CD38 axis is part of a network of accessory signals that modify the microenvironment, favoring localization of MM cells to growth-permissive sites (CD31 is also expressed by endothelial cells). Collectively, the authors finding might hold a theragnostic potential for CD38 targeting and overcoming drug resistance, potentially, these findings may point towards a potential Achilles’ heel of multiple myeloma that might be exploited therapeutically in the future (please refer to PMID: 31936617 and expand introduction/discussion section).

6. PLOS authors have the option to publish the peer review history of their article (what does this mean?). If published, this will include your full peer review and any attached files.

Reviewer #1: No

Reviewer #2: No

---

## [Author Response · Author response to Decision Letter 0]

15 May 2022

Please see the cover letter and detailed response to reviewer comments. Thanks.

---

## [Decision Letter · Decision Letter 1]

20 Jun 2022

LTβR overexpression promotes plasma cell accumulation

PONE-D-22-01475R1

Dear Dr. Cyster,

We’re pleased to inform you that your manuscript has been judged scientifically suitable for publication and will be formally accepted for publication once it meets all outstanding technical requirements.

Kind regards,

Antonio Solimando

Academic Editor

PLOS ONE

Additional Editor Comments (optional):

The authors have clarified several of the questions raised in the previous reviews. Most of the major problems have been addressed by this revision. The manuscript is considered acceptable for publication.

Reviewers' comments:

Reviewer's Responses to Questions

**Comments to the Author**

1. If the authors have adequately addressed your comments raised in a previous round of review and you feel that this manuscript is now acceptable for publication, you may indicate that here to bypass the “Comments to the Author” section, enter your conflict of interest statement in the “Confidential to Editor” section, and submit your "Accept" recommendation.

Reviewer #1: All comments have been addressed

Reviewer #2: All comments have been addressed

2. Is the manuscript technically sound, and do the data support the conclusions?

Reviewer #1: Yes

Reviewer #2: Yes

3. Has the statistical analysis been performed appropriately and rigorously? 

Reviewer #1: Yes

Reviewer #2: Yes

4. Have the authors made all data underlying the findings in their manuscript fully available?

Reviewer #1: Yes

Reviewer #2: Yes

5. Is the manuscript presented in an intelligible fashion and written in standard English?

Reviewer #1: Yes

Reviewer #2: Yes

6. Review Comments to the Author

Reviewer #1: The authors have clarified several of the questions I raised in my previous review. Most of the major problems have been addressed by this revision.

Reviewer #2: I am satisfied with the answers provided. The authors provided complete explanations and answers to the major issues.

7. PLOS authors have the option to publish the peer review history of their article (what does this mean?). If published, this will include your full peer review and any attached files.

Reviewer #1: No

Reviewer #2: No

---

## [Editor Report · Acceptance letter]

25 Jul 2022

PONE-D-22-01475R1 

LTβR overexpression promotes plasma cell accumulation 

Dear Dr. Cyster:

I'm pleased to inform you that your manuscript has been deemed suitable for publication in PLOS ONE. Congratulations! Your manuscript is now with our production department. 

Kind regards, 

on behalf of

Dr. Antonio Solimando 

Academic Editor

PLOS ONE